# Haploinsufficiency by minute *MutL homolog 1* promoter DNA methylation may represent unique phenotypes of microsatellite instability-gastric carcinogenesis

Hiroki Harada[1], Yusuke Nie[2], Ippeita Araki[1], Takafumi Soeno[1], Motohiro Chuman[1], Marie Washio[1], Mikiko Sakuraya[1], Hideki Ushiku[1], Masahiro Niihara[1], Kei Hosoda[1], Yusuke Kumamoto[2], Takeshi Naitoh[3], Takafumi Sangai[4], Naoki Hiki[1], Keishi Yamashita[1,5]*

1 Department of Upper-gastrointestinal Surgery, Kitasato University School of Medicine, Sagamihara, Kanagawa, Japan, 2 Department of General, Pediatric and Hepatobiliary-Pancreatic Surgery, Kitasato University School of Medicine, Sagamihara, Kanagawa, Japan, 3 Department of Lower Gastrointestinal Surgery, Kitasato University School of Medicine, Sagamihara, Kanagawa, Japan, 4 Department of Breast and Thyroid Surgery, Kitasato University School of Medicine, Sagamihara, Kanagawa, Japan, 5 Division of Advanced Surgical Oncology, Department of Research and Development Center for New Medical Frontiers, Kitasato University School of Medicine, Sagamihara, Kanagawa, Japan

* keishi23@med.kitasato-u.ac.jp

**Data Availability Statement:** All relevant data are within the paper and its Supporting Information files.

## Abstract

Promoter DNA methylation of *MutL homolog 1 (MLH1)* is considered to play a causative role in microsatellite instability (MSI) carcinogenesis in primary gastric cancer, and a high MSI status is associated with treatment sensitivity to human cancers. Nevertheless, clinico-pathological analysis is defective for *MLH1* methylation status in a quantitative manner. We newly developed quantitative methylation specific PCR using a TaqMan probe and applied it to 138 patients with primary gastric cancer who underwent gastrectomy in addition to basic molecular features such as MSI, Epstein Barr virus, and other DNA methylation status. (1) In primary gastric cancer, median methylation value was 0.055, ranging from 0 to 124.3. First, *MLH1* hypermethylation was strongly correlated with MSI-High/MSI-Low status and suppressed immunostaining ($P < 0.0001$). (2) The *MLH1* hypermethylation was associated with advanced age ($P = 0.0048$), antral location ($P = 0.0486$), synchronous multiple gastric cancer ($P = 0.0001$), and differentiated histology ($P = 0.028$). (3) Log-rank plot analysis identified the most relevant cut-off value (0.23) to reflect gentle phenotypes in *MLH1* hypermethylation cases ($P = 0.0019$), especially in advanced gastric cancer ($P = 0.0132$), which are designated as haploinsufficiency of MSI (MSI-haplo) phenotype in this study. (4) In synchronous multiple gastric cancer, *MLH1* hypermethylation was not necessarily confirmed as field cancerization. (5) MSI-haplo defined by *MLH1* methylation status represented distinct prognostic phenotype even after molecular classifications. *MLH1* hypermethylation designated as MSI-haplo may represent unique prognostic phenotype during gastric carcinogenesis.

**Funding:** The authors received no specific funding for this work.

**Competing interests:** The authors have declared that no competing interests exist.

## Introduction

Gastric cancer is the fifth most common malignancy and the third leading cause of cancer-related deaths worldwide [1]. Cancer is a genetic disease, and cancer cell survival is addicted to the specific oncogenes aberrated in tumor cells like *HER2* in gastric cancer [2, 3]. Therefore, sophisticated molecular understanding is critical for determining therapeutic targets of gastric cancer.

A comprehensive molecular classification of gastric cancer was proposed as Epstein Barr virus (EBV)-associated (8.8%), microsatellite instability (MSI)-associated (21.7%), chromosomal instable (49.8%), and genomic stable (19.0%) gastric cancer [4]. Among these four definite molecular phenotypes, MSI-associated gastric cancer representing defective DNA mismatch repair system is intriguing, because it could be a predictive biomarker for immune checkpoint inhibitors [5–7] in addition to classical chemotherapy [8]. It has been reported that the clinicopathological factors in the MSI-associated gastric cancer were elderly, female, intestinal type in Lauren classification, and tumors located in the middle to lower site of the stomach [4]. Moreover, recently, MSI-associated cancers harbored frequent genomic mutations of an SWI/SNF chromatin-remodeling factor such as ARID1A [9, 10], and such tumor cells are a good candidate for synthetic lethal target for glutathione metabolism [11].

Nevertheless, genomic mutation of DNA mismatch repair genes has rarely been documented in sporadic gastric carcinoma with MSI-High (MSI-H) [10], and several early papers reported that hypermethylation of the CpG island in the *MutL homolog 1 (MLH1)* promoter DNA presents with a loss of MLH1 protein expression in almost all cases of MSI-H sporadic gastric carcinomas [10, 12–14]. These critical findings suggested that MSI-H is largely caused by *MLH1* hypermethylation in sporadic gastric cancer tumor tissues.

Most of the data on *MLH1* hypermethylation were evaluated by non-quantitative methylation-specific PCR (MSP). In previous similar reports evaluated by quantitative MSP, the quantified methylation values were not applied to investigate clinicopathological factors. Thus, there have been no reports describing the exact clinicopathological significance of *MLH1* methylation status in human cancers in a quantitative assessment. Allowing for the re-emerging importance of the MSI-H phenotype in gastric cancer clinics, it is time that *MLH1* methylation status is assessed by a very accurate methylation analysis.

In this paper, we newly developed a DNA methylation quantification system for *MLH1* methylation in quantitative MSP, and for the first time clarified the detailed clinicopathological features of *MLH1* methylation in primary gastric cancer.

## Materials and methods

### Cell lines

The colorectal cancer (CRC) cell lines SW48 and SW480 were used for positive and negative control for the *MLH1* methylation, respectively. SW48 and SW480 cells were purchased from the American Type Culture Collection (Manassas, VA, USA). SW48 and SW480 cells were grown in RPMI 1640 medium (GIBCO, Carlsbad, CA) containing 10% fetal bovine serum. The six gastric cancer cell lines (Kato III, KE-97, MKN74, NUGC-4, SH-10-TC, and MKN7) were recently used similarly and described [15].

### Patients and tissue samples

This study investigated 138 patients who underwent surgical resection for primary gastric cancer at the Kitasato university hospital, Japan in 2005 as described [15]. We extracted DNA from 138 tumor tissues. These tissue samples were collected from all patients who provided

written informed consent for the use of their pathological specimens. The present study was approved by the Ethics Committee of Kitasato University (Number B18-058). Patient characteristics are shown in S1 Table. Patients with neoadjuvant chemotherapy were not included in this study.

## DNA extraction and sodium bisulfite conversion

Tissue sections from primary tumors were stained with hematoxylin and eosin, and dissected under a microscope. Genomic DNA was extracted from formalin-fixed paraffin embedded (FFPE) tissues using a QIAamp DNA FFPE Tissue Kit (QIAGEN Sciences, Hilden, Germany). Bisulfite treatment was done according to the manufacturer's instructions of an EZ DNA Methylation-GoldTM Kit (Zymo Research, Orange, CA) as described [15].

## Direct sequence and clones' sequence for methylation positive control SW48 cells and negative control SW480 cells

We originally designed bisulfite sequence primers in this study as shown in Fig 1A. The primer sequences used in this manuscript are also included in S2 Table. The PCR products were separated on 1.5–2.0% agarose gel, then visualized them by ethidium bromide staining prior to direct sequence. Distilled water was used as negative control. Cloned sequence was done for 10 clones each in SW48 and SW480 cells, respectively as previously described [16].

## qMSP

qMSP was carried out using iQ Supermix (Bio-Rad Laboratories, Hercules, CA) in triplicate on the C1000 TouchTM Thermal Cycler CFX96 Real Time System (Bio-Rad) as described [15]. Bisulfite-treated DNA was amplified by the following PCR conditions: 95˚C for 10 min, followed by 40 cycles at 95˚C for 15 sec, annealing temperature (60˚C) for 1 min.

The sequence primers and probes for quantitative assessment were designed with early references regarding conventional methylation-specific PCR [17, 18] (Fig 1A) according to a primer design software (MethPrimer in the National Center for Biotechnology Information-NCBI home page). Our unique combination of primers and probe for qMSP is the most similar with that belonging to Shibata [19], which did not investigate clinicopathological information.

Serial dilutions of bisulfite modified DNA from the human colon carcinoma cell line SW48 was used to construct the calibration curve on each plate as a methylation positive control, and the human colon carcinoma cell line SW480 was used as a negative control. The methylation value was defined as a TaqMeth value by the ratio of the amplified signal value of methylated *MLH1* to the value for *β-actin*, which was then multiplied by 100.

## Immunohistochemistry

Immunostaining was performed on formalin-fixed, paraffin-embedded sections (4 μm thick). Sections were incubated using the anti-MLH1 mouse monoclonal antibody (dilution of 1:50) (BD Pharmingen, NJ, US). Immune complexes were detected with a Histofine Simple Stain MAX PO (MULTI) (Nichirei, Tokyo, Japan), following the manufacturer's protocol, and visualized using the 3,3'-diaminobenzidine (DAB) substrate. Sections were counter-stained with Hematoxylin solution.

## PCR amplification of EBV DNA and in situ hybridization (ISH)

For EBV infection, *EBV nuclear antigen 1 (EBNA1)* DNA was used in this study. Primers and probes of *EBNA1* were synthesized as published sequences [20]. Real-time PCR for both

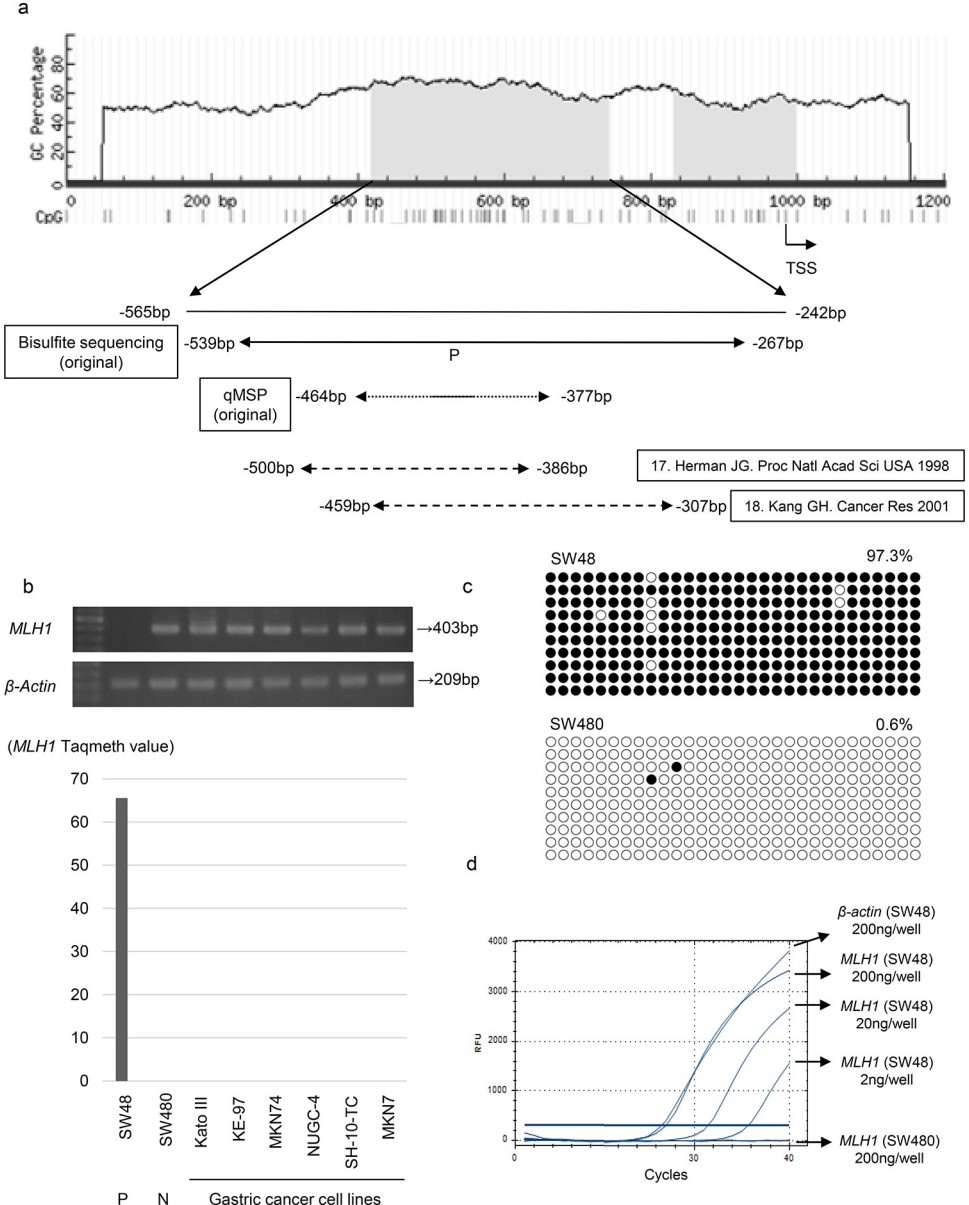

**Fig 1. Analysis of *MLH1* promoter methylation and expression in gastric cancer cell line and colon cancer cell line.** (a) CpG islands (gray area) in the 5′-flanking regions of *MLH1* (bottom). Solid and dotted lines below the bottom panel indicate the sequences for bisulfite sequencing analysis or quantitative methylation specific PCR (qMSP), respectively. Primer design in early references regarding conventional methylation-specific PCR (reference number 17, 18). The bold line in qMSP (original) indicates the location of the Taqman probe. TSS, translation start site. (b) RT-PCR and qMSP showed silenced expression in SW48 (positive control) with *MLH1* hypermethylation, whereas SW480 (negative control) without *MLH1* hypermethylation showed definite expression. In addition, RT-PCR proved to be expressed for *MLH1* in all 6 gastric cancer cell lines, indicating that no gastric cancer cell lines we have are expected to harbor *MLH1* hypermethylation. (c) Methylation status of 30 individual CpG sites (columns) of 10 cloned PCR products (rows), within the *MLH1* promoter region by bisulfite sequencing. Open and filled circles, unmethylated and methylated CpG sites, respectively. Numbers denote qMSP value. Results shown for SW48 and SW480 served as positive and negative control. (d) PCR amplification of methylated *MLH1* and *β-actin*.

*EBNA1* and *β-actin* were performed using iQ Supermix (Bio-Rad Laboratories, Hercules, CA) in triplicate on the C1000 Touch ™ Thermal Cycler CFX96 Real Time System (Bio-Rad Laboratories, Hercules, CA). The thermocycling conditions were as follows: 50˚C for 2 min, 95˚C

for 10 min, 95˚C for 15 sec and 60˚C for 1 min for 40 cycles. The relative amounts of *EBNA1* DNA compared to those of *β-actin* was calculated as follows: Ct (*EBNA1*)-Ct (*β-actin*) for each sample. Relative copy number was determined as 2-ΔΔCt, where ΔΔCt = ΔCt (tumor)- ΔCt (corresponding normal) [21]. An increase in gene copy number was considered when 2-ΔΔCt was 1.0 or more.

EBV encoded small RNA (EBER)-ISH were performed using the Ventana BenchMark XT automatic immunostainer following the manufacturer's recommendations (Ventana Medical Systems, Tucson, AZ, USA) in the recent 2 cases with EBV positive as a routine test, and also assessed DNA copy numbers. Protein removal and nucleic acid exposures used ISH protease 1. The EBER probe of Ventana DNP labeled probes was then applied and hybridized. To detect signals, antibodies (CS1, CS2, CS3, CS4) were employed. The reaction was done with ISH iVIEW BLUE-Research Kit.

### MSI procedures

MSI analysis was done by evaluating two mononucleotides repeat markers (BAT25 and BAT26) and three dinucleotide repeat markers (D2S123, D5S346, and D17S250) as recommended by the National Cancer Institute workshop for MSI [22]. Each marker was PCR-amplified in a separate 10 μL reaction containing 5 μL Type-it Microsatellite PCR kit (QIAGEN Sciences, Hilden, Germany), 10 pmol each of labeled forward and unlabeled reverse primers, and 10 ng of template DNA. D2S123 and BAT26 were labeled with 6-FAM, BAT25 with VIC, D5S346 with NED, and D17S250 with PET. PCR was done in a Thermal Cycler GeneAtlas S (ASTEC, Fukuoka, Japan).

The reaction conditions were as follows: initial denaturation at 95˚C for 5 min, 28 cycles of (95˚C for 30 sec, 57˚C for 90 sec, 72˚C for 30 sec), followed by a final extension at 60˚C for 30 min. Approximately 1 to 2 ng of each PCR product were mixed with a 1:40 dilution of Hi-Di Formamide-GeneScan LIZ500 Size Standard and visualized on a Genetic Analyzer 3500xL (Applied Biosystems, Foster City, CA, USA). Allelic size alterations were detected using Gene-Mapper Software Version 4.1 (Applied Biosystems, Foster City, CA, USA).

Samples were considered positive for MSI when alternate-sized bands were present in the tumor DNA but absent in the respective control mucosal DNA. Tumors were classified as MSI-H if ≥2 markers had allelic shifts, low level MSI (MSI-L) if only one of the five markers had allelic shift, and microsatellite-stable (MS-stable) when no marker showed allelic shift [22].

### Statistical analysis

Continuous variables were evaluated by ANOVA, Student's t test; categorical variables by Fisher's exact test or the Chi-square test, as appropriate. Clinicopathological characteristics and follow up data were evaluated in terms of overall survival (OS). The follow-up time was calculated from the date of surgery to death or end-point. OS was estimated by the Kaplan-Meier method, and compared using the log-rank test. Variables suggesting potential prognostic factors on univariate analyses were subjected to a multivariate analysis using a Cox proportional-hazards model. A P-value <0.05 was considered to indicate statistical significance. All statistical analyses were conducted using the SAS software package (JMP Pro11, SAS Institute, Cary, NC).

## Results

### *MLH1* promoter DNA hypermethylation in control cell lines

We initially used CRC cell lines, SW48 and SW480 cells, as positive and negative controls for methylation assessment as previously shown in conventional MSP [17]. As expected, SW48

cells with *MLH1* hypermethylation were accompanied by its silenced expression, while SW480 cells with no *MLH1* hypermethylation exhibited its definite expression (Fig 1B). RT-PCR proved to be clearly expressed for *MLH1* in all six gastric cancer cell lines (Fig 1B, upper panel), indicating that no gastric cancer cell lines harbors *MLH1* hypermethylation.

The degree of *MLH1* promoter hypermethylation by cloned sequences has not been determined anywhere even in the control cells. Using bisulfite treatment of DNA followed by direct sequence (S1 Fig) and cloned sequence analysis, we clarified the methylation level of the *MLH1* promoter sequence in SW48 cells (methylation positive control) and SW480 cells (methylation negative control) (Fig 1C). Consequently, we found methylated CpG sites in 99.3% and 0.6% of the promoter DNA in SW48 and SW480 cells, respectively (Fig 1C). These findings suggested that SW48 cells and SW480 cells could be designated as positive and negative controls in our quantitative assessment, respectively.

We subsequently quantified methylated *MLH1* after bisulfite DNA treatment using qMSP with primers and probe originally designed for this current study (Fig 1A). The efficacy of PCR amplification of methylated *MLH1* was excellent and found to be comparable with that of *β-actin* (Fig 1D). *MLH1* hypermethylation could be clearly detected for DNA from SW48 cells at concentrations of 1-fold (200 ng/well), 10-fold (20 ng/well), 100-fold (2 ng/well) dilutions, but could not be detected for those of 1000-fold (0.2 ng/well), and 10000-fold (0.02 ng/well) dilutions, whereas no methylation at all could be detected in negative control SW480 DNA templates (200 ng, 20 ng, 2 ng, 0.2 ng, 0.02 ng/well) (Fig 1D). We used fluorescence of 350 RFU (relative fluorescence unit) as the threshold line in this assay, because the negative control is always below this threshold. These findings suggested that even dense *MLH1* methylation such as that found in SW48 cells can barely be detected using qMSP when the DNA is diluted beyond 1000-fold.

Using this optimal condition, *MLH1* methylation was never seen in six gastric cancer cell lines, which were abundantly expressed for *MLH1* mRNA (Fig 1B, lower panel). Hence, promoter DNA methylation status is completely consistent with mRNA expression status of the *MLH1* in the six gastric cancer cells as well as the two CRC cell lines.

## *MLH1* promotor DNA methylation and its correlation with clinicopathological factors in primary gastric cancer

On hundred and thirty-eight primary tumor specimens of patients with gastric cancer who underwent surgical resection were assessed by qMSP to evaluate the clinical relevance of the *MLH1* methylation status. The median TaqMeth value was 0.055, ranging from 0 to 124.3 in the 138 tumor tissues (T) (Fig 2A).

The correlation of the *MLH1* TaqMeth value to clinicopathological factors was initially evaluated by ANOVA in primary gastric cancer. The negative prognostic factors (e.g., staging factors) showed no significant association with *MLH1* TaqMeth value by ANOVA (Fig 2B). Conversely, there was significant correlation of a high *MLH1* TaqMeth value to synchronous gastric cancer ($P = 0.0001$, Fig 2C), advanced age ($P = 0.0048$, Fig 2D), histological type ($P = 0.028$, Fig 2E), and antral location of the tumor ($P = 0.049$) (Fig 2F).

As the *MLH1* TaqMeth value was the most strongly associated with synchronous gastric cancer, we then investigated synchronous other tumor tissues together with the corresponding non-cancerous gastric mucosa tissues (Fig 2G). Surprisingly, field cancerization of the double cancer representing *MLH1* hypermethylation of the corresponding non-cancerous tissues was unexpectedly limited to only a few cases (6/17 cases, 35%). Threshold was not set to discriminate the corresponding normal mucosa from tumor tissues.

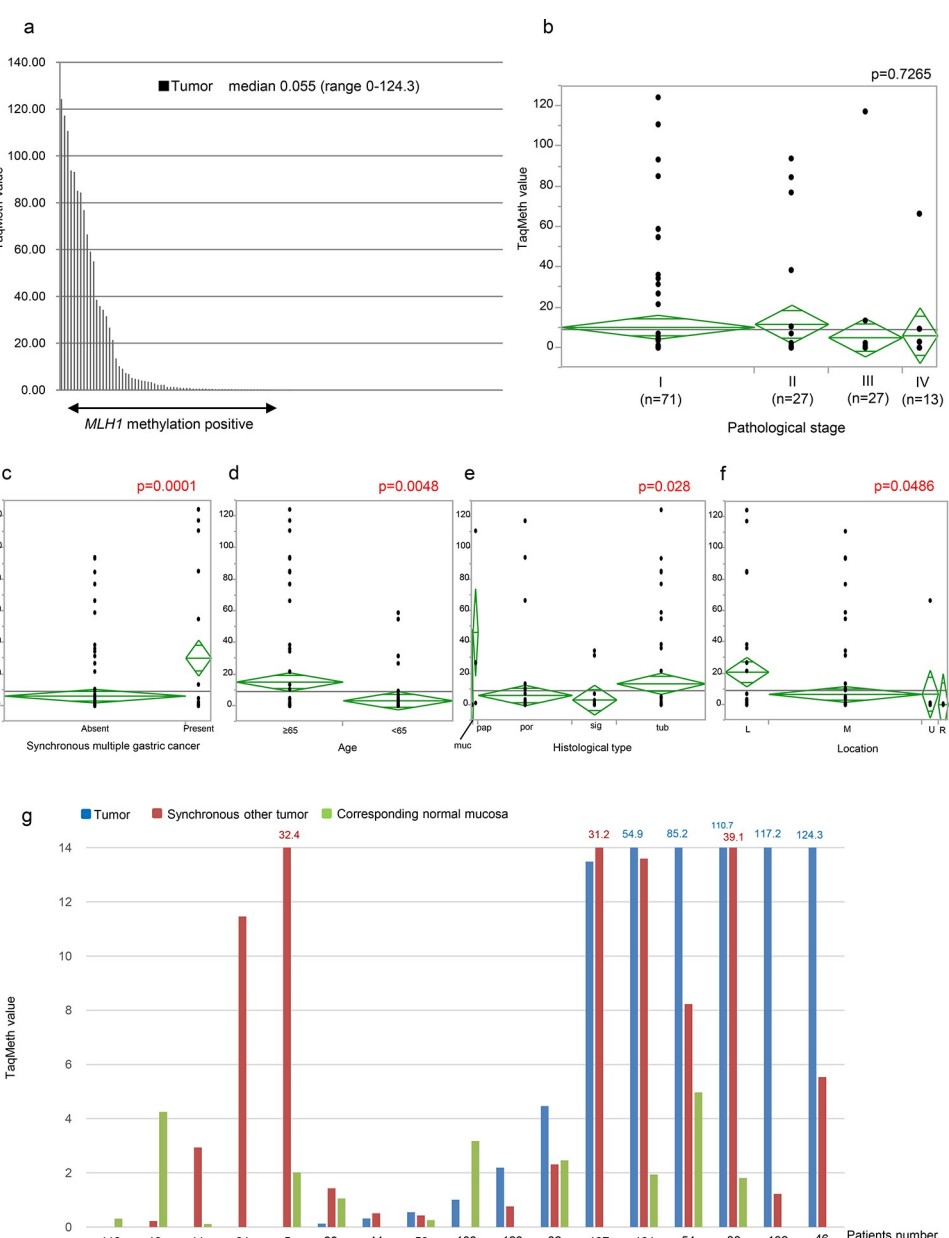

**Fig 2. *MLH1* TaqMeth value in primary gastric cancer and its correlation with clinicopathological factor.** (a)
*MLH1* TaqMeth value distribution in the gastric cancer tissues. Median TaqMeth value of *MLH1* was 0.055, ranging
from 0 to 124.3. *MLH1* TaqMeth value distribution in the gastric cancer tissues according to (b) Pathological stage, (c)
Synchronous multiple gastric cancer, (d) Age, (e) Histological type, (f) Tumor location. (g) *MLH1* TaqMeth value in
the corresponding non-cancerous gastric mucosa tissues, cancerous tissue and synchronous other tumor tissue.
Abbreviation: muc; Mucinous adenocarcinoma, pap; Papillary adenocarcinoma, por; Poorly differentiated
adenocarcinoma, sig; Signet-ring cell carcinoma, tub; Tubular adenocarcinoma, L; Lower site, M; Middle site, U;
Upper site, R; Residual gastric cancer.

## Multivariate prognostic analysis in primary gastric cancer

We further investigated the correlation of the *MLH1* TaqMeth value to prognosis (OS) in primary gastric cancer. To determine the optimal cut-off values for predicting poor prognosis, we
assessed each p-value and relative risk by the log rank plot analysis as previously described

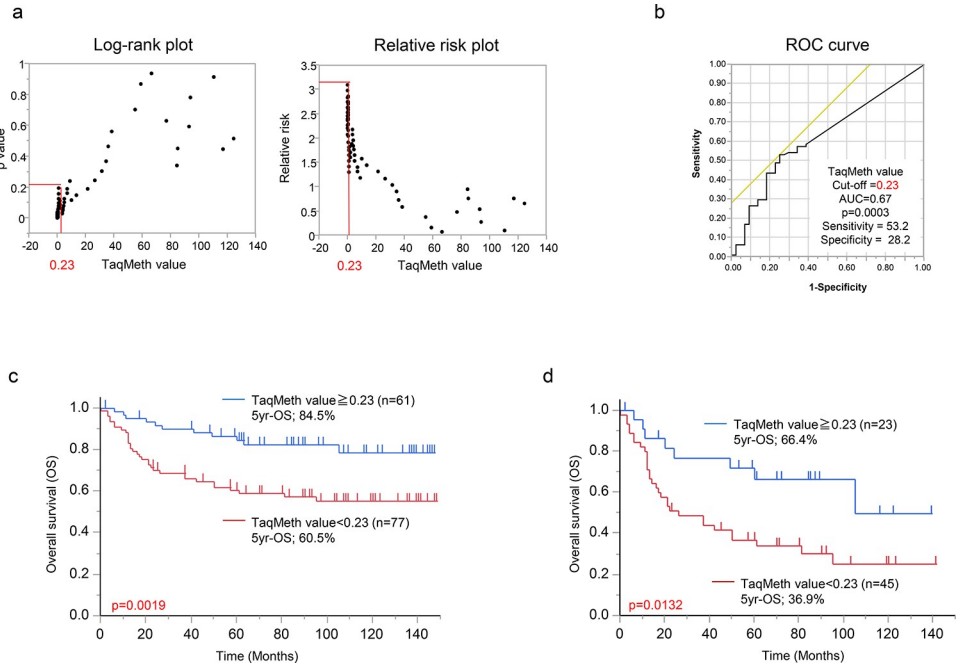

**Fig 3. Prognostic analysis of *MLH1* TaqMeth value in primary gastric cancer.** (a) Log-rank plot analysis of the optimal cutoff value of *MLH1* TaqMeth value in death event. (b) Receiver operating characteristics (ROC) curve to predict deaths also recapitulated the same optimal cut-off value of Log-rank plot analysis. (c) Kaplan-Meier survival curves for overall survival (OS) comparing gastric cancer patients with *MLH1* TaqMeth value below 0.23 and those with *MLH1* TaqMeth value equal to or over 0.23 ($P = 0.0019$). (d) Kaplan-Meier survival curves for OS comparing gastric cancer patients with *MLH1* TaqMeth value 0.23 in advanced gastric cancer patients (depth of invasion of pT2 or beyond), ($P = 0.0132$).

[15]. The most optimal cut-off value was defined as 0.23 with the highest relative risk in primary gastric cancer (Fig 3A). Intriguingly, receiver operating characteristics curve to predict deaths also recapitulated the same optimal cut-off value of 0.23 (Fig 3B, area under curve = 0.67). These findings suggested that this cut-off value of *MLH1* methylation has the most optimally prognostic relevance in primary gastric cancer.

Univariate prognostic analysis showed that age ($P = 0.0064$), morphological types ($P < 0.0001$), pathological stage ($P < 0.0001$), postoperative complication ($P = 0.0034$), tumor resectability ($P < 0.0001$), and the *MLH1* TaqMeth value were significantly associated with prognosis in primary gastric cancer (S3 Table). The identified univariate prognostic factors were subjected to a multivariate Cox hazards model. Multivariate analysis only identified pathological stage as an independent prognostic factor in gastric cancer, and the *MLH1* TaqMeth value was not finally remnant (S3 Table).

We then determined the correlation between clinicopathological factors and *MLH1* methylation status using the most optimal cut-off value (0.23) by the Chi-square test in primary gastric cancer. Consequently, *MLH1* hypermethylation was significantly associated with tumor location ($P = 0.0298$), morphological type ($P = 0.0155$), depth of invasion ($P = 0.0233$), lymph node metastasis ($P = 0.0208$), pathological stage ($P = 0.0253$), and histological type ($P = 0.0024$). Multivariate regression analysis elucidated that *MLH1* hypermethylation is independently associated with lower portion of the tumor location ($P = 0.0092$) and differentiated histology ($P = 0.0072$). These findings suggested that the *MLH1* TaqMeth value was not independent prognostic factors due to close correlation to both tumor location and differentiated histology (Table 1).

**Table 1. Correlation between clinicopathological factors and *MLH1* methylation status divided by cutoff value of 0.23 in primary gastric cancer.**

| Variable | MLH1 methylation | | | Multivariate analysis | | |
|---|---|---|---|---|---|---|
| | ≥0.23 (n = 61) | <0.23 (n = 77) | *p*-value | Odds ratio | 95% CI | *p*-Value |
| Gender | | | 0.9635 | | | |
| Male | 37 (60.7%) | 47 (61.0%) | | | | |
| Female | 24 (39.3%) | 30 (39.0%) | | | | |
| Age (years) | | | 0.3131 | | | |
| <65 | 28 (45.9%) | 42 (54.6%) | | | | |
| ≥65 | 33 (54.1%) | 35 (45.4%) | | | | |
| Tumor location | | | 0.0298 | | | |
| Middle | 39 (63.9%) | 60 (77.9%) | | Reference | | |
| Upper | 4 (6.6%) | 8 (10.4%) | | 1.34 | 0.32–5.64 | 0.6891 |
| Lower | 18 (29.5%) | 9 (11.7%) | | 3.7 | 1.38–9.9 | 0.0092 |
| Morphological type | | | 0.0155 | | | |
| Early type | 38 (62.3%) | 32 (41.6%) | | | | |
| Advanced type | 23 (37.7%) | 45 (58.4%) | | | | |
| Synchronous multiple gastric cancer | | | 0.0691 | | | |
| Absence | 50 (82.0%) | 71 (92.2%) | | | | |
| Presence | 11 (18.0%) | 6 (7.8%) | | | | |
| Depth of tumor invasion | | | 0.0233 | | | |
| Superficial invasion | 38 (62.3%) | 33 (42.9%) | | | | |
| Advanced invasion | 23 (37.7%) | 44 (57.1%) | | | | |
| Lymph node metastasis | | | 0.0208 | | | |
| Absence | 39 (63.9%) | 34 (44.2%) | | | | |
| Presence | 22 (36.1%) | 43 (55.8%) | | | | |
| Peritoneal lavage cytology (CY) | | | 0.1143 | | | |
| CYX | 21 (34.4%) | 27 (35.0%) | | | | |
| CY0 | 38 (62.3%) | 40 (52.0%) | | | | |
| CY1 | 2 (3.3%) | 10 (13.0%) | | | | |
| Peritoneal dissemination (P) | | | 0.3927 | | | |
| P0 | 59 (96.7%) | 72 (93.5%) | | | | |
| P1 | 2 (3.3%) | 5 (6.5%) | | | | |
| Pathological stage (pStage) | | | 0.0253 | | | |
| pStage I | 39 (63.9%) | 32 (41.5%) | | | | |
| pStage II | 12 (19.7%) | 15 (19.5%) | | | | |
| pStage III | 7 (11.5%) | 20 (26.0%) | | | | |
| pStage IV | 3 (4.9%) | 10 (13.0%) | | | | |
| Histological type | | | 0.0024 | | | |
| Undifferentiated | 28 (45.9%) | 55 (71.4%) | | Reference | | |
| Differentiated | 33 (54.1%) | 22 (28.6%) | | 2.92 | 1.33–6.39 | 0.0075 |

Patients with gastric cancer with *MLH1* hypermethylation (≥ 0.23) showed better prognosis than those with *MLH1* hypomethylation (< 0.23) (5-year OS: 84.5% vs 60.5%, *P* = 0.0019) (Fig 3C). This prognostic tendency is recognized, especially in advanced (depth of invasion of pT2 or beyond) gastric cancer (*P* = 0.0132) (Fig 3D), but neither in early (depth of invasion of pT1) gastric cancer (*P* = 0.7904) nor in recurrent cases undergoing chemotherapy (*P* = 0.5712) (S2 Fig). In curative surgery, gastric cancer with *MLH1* hypermethylation had 7 recurrences (1 liver, 1 lung, 2 bone, 1 brain, 1 adrenal gland, 1 peritoneum, in which less peritoneal dissemination was seen than in *MLH1* hypomethylation cases).

## Promoter DNA methylation of *MLH1* critically affects its protein expression in tumor tissues

MLH1 protein expression was immune-stained in the 10 gastric cancer tumor tissues with the highest and the lowest values of the methylation using anti-MLH1 polyclonal antibody (Fig 4A). A strong expression of MLH1 protein was observed in tumor tissues with *MLH1* hypomethylation, whereas a weak expression was dominant in those with *MLH1* hypermethylation. The differential expression of the MLH1 protein categorized into 3 groups (0, 1+, 2+) showed significantly different *MLH1* TaqMeth values ($P < 0.0001$) (Fig 4B), suggesting that MLH1 protein expression is significantly correlated with the *MLH1* hypermethylation.

## *MLH1* hypermethylation and MSI-associated gastric cancer in the context of molecular features

We then compared the *MLH1* TaqMeth values with MSI status. In the 136 gastric cancer samples, MSI-H was confirmed in 12 (8.8%), MSI-L in 5 (3.7%), and MSS in 119 (87.5%) (Fig 5A). To predict MSI-H/MSI-L, the most cut-off value of *MLH1* was 2.23 (Fig 5B), where we found *MLH1* hypermethylation in 91.7% of MSI-H (11/12), 80% of MSI-L (4/5), and 13.4% of MSS (16/119), respectively (Fig 5C). Conversely, to predict MSI-H, the most optimal cut-off value of *MLH1* was 38.55 (Fig 5D), where *MLH1* hypermethylation was seen in 83.3% of MSI-H (10/12), 0% of MSI-L (0/5), and 1.7% of MSS (2/119), respectively (Fig 5E). These findings suggested that MSI status is reflected by *MLH1* methylation degree. Thus, in this study, we will designate the 3 kinds of MSI status MSI-H, MSI-L, and haploinsufficiency of MSI (MSI-haplo) which could be delineated by *MLH1* methylation values (Fig 5F).

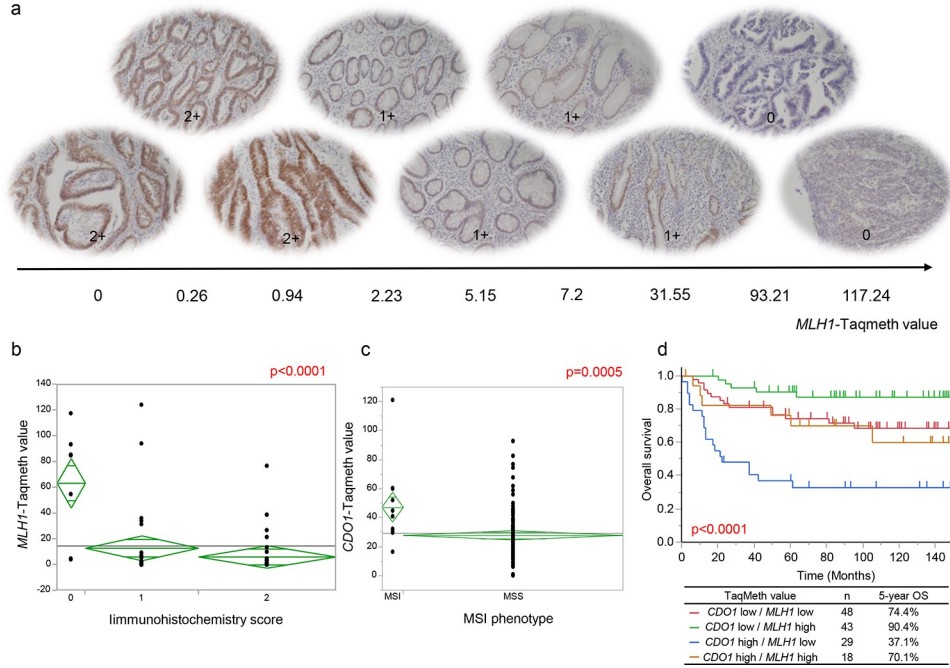

**Fig 4.** (a) *MLH1* promoter DNA methylation affecting protein expression in tumor tissue. Expression of MLH1 proteins in three groups (0, 1 +, 2 +) with different *MLH1*TaqMeth value. (b) *MLH1* TaqMeth value distribution in the gastric cancer tissues according to three groups of immunochemistry score (0, 1+, 2+). (c) *CDO1* TaqMeth value distribution in the gastric cancer tissues according to microsatellite instability (MSI) phenotype (MSI or Microsatellite stable; MSS). (d) Kaplan-Meier survival curves for overall survival comparing gastric cancer patients with combination of *MLH1* and *CDO1* TaqMeth value ($P < 0.0001$).

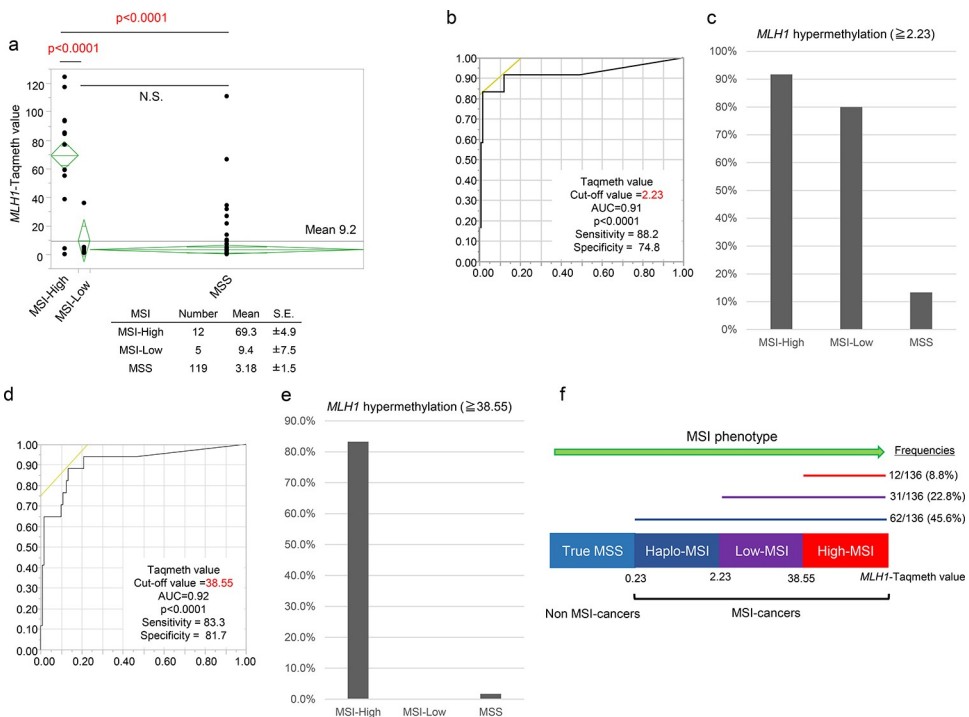

**Fig 5.** (a) *MLH1* TaqMeth value distribution in the gastric cancer tissues according to three groups of microsatellite instability (MSI) (MSI-High, MSI-Low, Microsatellite stable; MSS). (b) Receiver operating characteristics (ROC) of the optimal cutoff value of *MLH1* TaqMeth value in MSI-High and Low. (c) Percentage of MSI-High, MSI-Low, MSS equal to or over *MLH1* TaqMeth value 2.23. (d) ROC of the optimal cutoff value of *MLH1* TaqMeth value in MSI-High. (e) Percentage of MSI-High, MSI-Low, MSS equal to or over *MLH1* TaqMeth value 38.55. (f) The four kinds of MSI phenotype (MSI-H, MSI-L, and haploinsufficiency of MSI; MSI-haplo, MSS) which could be delineated by *MLH1* methylation values. Optimal cut off value of *MLH1* TaqMeth value and its frequency for each MSI phenotype.

As the *MLH1* methylation value increased in the order of True MSS, Haplo-MSI, Low-MSI, and High MSI, the gastric cancers with lower tumor site (*P* = 0.0313), the synchronous gastric cancers (*P* = 0.0122) and the histologically differentiated types (*P* = 0.0052) increased (S4 Table).

Recent molecular classification of gastric cancer is categorized into EBV-associated, MSI-associated, and other gastric cancers [4]. In this study, we initially detected EBV-associated gastric cancer by simple PCR assessment (Fig 6A, left panel) in gastric cancer cases which had been confirmed positive by ISH in the recent routine tests (representative ISH shown in Fig 6B). We also applied quantitative PCR to 138 cases (right panel of Fig 6A) as well as the 2 recent cases (Fig 6B) (designated as simple EBV positive, sEBV in this study). EBV positive gastric cancers were mutually exclusive from MSI-H/L as previously shown [4], and almost not redundant with *MLH1* methylation status (2 cases redundant in MSI-haplo) (Fig 6A, right panel).

This simple molecular classification revealed that the best prognosis was again shown in MSI-haplo (n = 59) in contrast to sEBV (n = 7), and other gastric cancers (n = 72) in Fig 6C (*P* = 0.0167), and MSI-haplo showed 83.9% OS in gastric cancer.

We recently demonstrated that *Cysteine dioxygenase type 1 (CDO1)* hypermethylation was cancer-specific in primary gastric cancer and its hypermethylation was correlated with poor prognosis in the same patient cohort [15]. As the CpG islands methylator phenotype (CIMP) including *MLH1* hypermethylation has been repeatedly proposed to be enriched in the specific

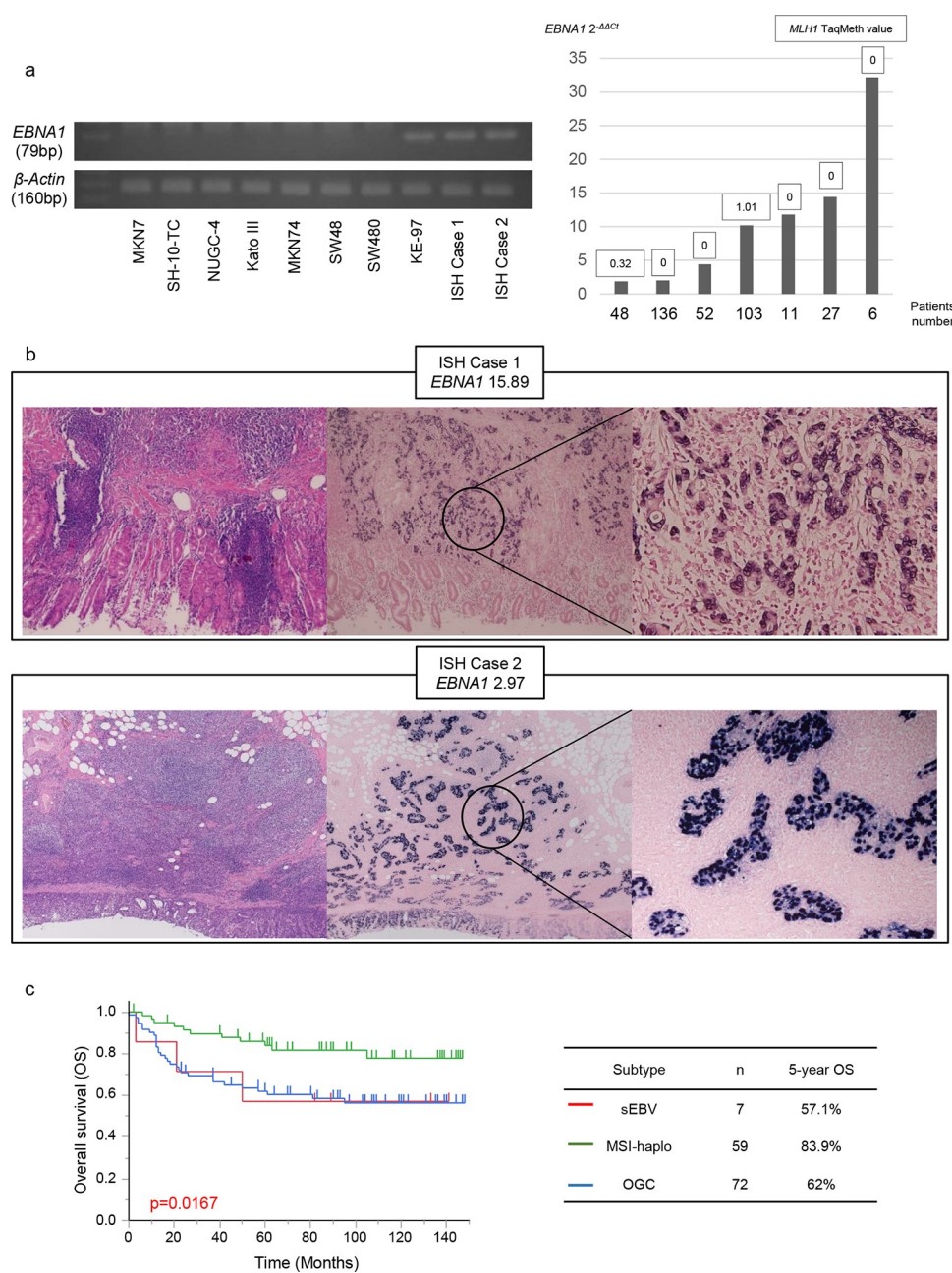

**Fig 6.** (a) Results of RT-PCR (left panel) using gastric cancer cell lines, colon cancer cell lines and in situ hybridization (ISH) positive cases in *EBNA1* and *β-actin*. Gastric cancer tissues with cases of equal to or over 1 in quantitative real-time PCR of *EBNA1*. *MLH1* TaqMeth values of these *EBNA1* positive cases (right panel). (b) Histological images of two cases confirmed positive for Epstein Barr virus (EBV) by ISH. Each *EBNA1* quantitative value. (c) Kaplan-Meier survival curves for overall survival comparing gastric cancer patients with MSI-haplo (n = 59), EBNA1 positive (n = 7) and other gastric cancers (OGC) (n = 72) (*P* = 0.0167).

patients [23], we finally compared *MLH1* methylation and *CDO1* methylation in gastric cancer. *MLH1* hypermethylation was significantly associated with *CDO1* hypermethylation (R = 0.433, *P* < 0.0001) (S3 Fig), and MSI status was also significantly correlated with *CDO1* hypermethylation like *MLH1* hypermethylation (Fig 4C), suggesting that MSI carcinogenesis may be associated with pan-epigenetic alterations. Combination of *MLH1* and *CDO1*

methylation status has important prognostic information in gastric cancer (Fig 4D), suggesting that prognosis is differentially affected by oncological function of the individual genes.

## Discussion

*MLH1* hypermethylation has been repeatedly examined so far by conventional MSP; however, quantitative analysis related to clinicopathological relevance is defective. The classical conventional MSP was judged by visual inspection, in which *MLH1* hypermethylation was recognized in 20–30% in gastric cancer and consistent with MSI-H [13, 24]. If the cut-off of the *MLH1* TaqMeth value was determined as the highest top 20% in our current study, it was corresponding to 3.5, which is close to that (2.3) to identify MSI-H/L phenotype (Fig 5D).

TaqMeth values of 0–5 in qMSP could be corresponding to visual detection by the inspection study [16], so our current data representing 2.3 of *MLH1* TaqMeth value may be consistent with the numerous previous literatures regarding *MLH1* methylation frequencies by the conventional MSP [13, 24].

Conversely, our current assessment clarified the best optimal cut-off value of 38.55 to identify MSI-H (Fig 5E), and such methylation level is consistent with silenced expression of MLH1 protein (Fig 4A). These findings, for the first time, proposed that *MLH1* methylation degree can affect MSI phenotype (MSI-H/MSI-L) in a stepwise manner. Our findings can also scientifically support the early report that MSI-H can develop from MSI-L or absence of MSI due to accumulation of DNA methylation during progression of early-stage gastric cancer [25].

From prognostic point of view, conversely, the best optimal cut-off value of the TaqMeth value of *MLH1* was 0.23 (designated as MSI-haplo-the top 45.6% in this study, Fig 5F), proposing a potential entity of MSI-haplo phenotype that shows good prognosis in our current study (Fig 3). This prognostic outcome is consistent with early reports exhibiting good prognosis in gastric cancer with conventional MSI-H or high *MLH1* methylation [26]. We are considering that haplo-MSI phenotype could be microsatellite-instable despite MSS defined by the current definition. The definition of MSI-H/L is based on the specific relevant markers (Bethesda panel), and they could not be covered with minor abnormalities of microsatellites caused by haplo-MSI that is corresponding to the lowest hypermethylation of *MLH1* ($> 0.23$) (Fig 5F). Prognostic stratification defined by the most optimal cut-off value (0.23, but not 2.3) might propose bona-fide entity of MSI phenotypes including haplo-MSI in gastric cancer. Hence, MSI-haplo phenotype determined by *MLH1* methylation in qMSP is still considered to sustain unique phenotypes of the conventional MSI from a prognostic point of view. We examined the methylation of *MLH1* in each MSI phenotype (MSI-H, MSI-L, and MSS) using a direct sequencing. It was considered that the sensitivity to detect methylation of direct sequencing is limited to *MLH1* hypermethylation cases, and MSI-haplo can only be detected by real-time PCR (S4 Fig).

Our data also showed that MSI-haplo was significantly associated with differentiated histology and antral location in a multivariate analysis (Table 1), and such phenotypes have been considered to be unique for conventional MSI phenotypes [4, 27–30]. Hence, MSI-haplo may be the newly emerging entity representing MSI carcinogenesis.

Among the clinicopathological features in our current study, *MLH1* hypermethylation was the most strongly associated with synchronous gastric cancer. This finding proposed *MLH1* involvement in field cancerization of gastric carcinogenesis described elsewhere [31]. In our data, however, *MLH1* was not necessarily hypermethylated in synchronous other gastric cancer, and/or the corresponding non-cancerous mucosal tissues (Fig 2F). Our current findings suggest that *MLH1* hypermethylation may be included among other epigenetic alterations among pan-epigenetic genes during gastric carcinogenesis.

We herein hypothesized field cancerization by multiple concurrent epigenetic alterations. Gastrointestinal cancer with the CIMP was proposed, which have included *MLH1* hypermethylation [32]. In fact, *MLH1* hypermethylation occurs predominantly in the setting of other gene hypermethylation such as *HPP1* reported from other group [19] and *CDO1* in the current study.

Conversely, *CDO1* hypermethylation was proven to be closely associated with *MLH1* hypermethylation (R = 0.433, *P* < 0.0001), and *CDO1* hypermethylation was also recognized specifically in MSH-H/L phenotypes (*P* = 0.0005, Fig 4C). These findings suggest that *MLH1* mediated MSI carcinogenesis is a part of pan-epigenetic carcinogenesis. We cannot explain field cancerization by *MLH1* hypermethylation alone.

MSI-H tumors have been reported to have marked infiltration of CD8 T cells [33, 34], suggesting that cancer immunity is highly activated. Tumor cells attacked by cancer immunity may be exposed to high regenerative oxygen species [35, 36], and the redox system including NF-E2-related factor 2 (NRF2) activation is required for cancer cell survival [37]. Tumor growth requires NRF2 without *CDO1* expression [38], so *CDO1* hypermethylation with its gene silencing was recognized in MSI-H tumors.

Conversely, *CDO1* hypermethylation is a prognostic marker of poor prognosis [15], while *MLH1* hypermethylation is a marker of good prognosis. In the current study, we clarified that *CDO1* hypermethylation with *MLH1* hypomethylation showed the worst prognosis in gastric cancer and vice versa (Fig 4D). The differential roles of the two discrete epigenetic genes in prognosis suggest that both genes do not necessarily share the common functional role in gastric carcinogenesis, although epigenetic mechanism is shared.

Smyth et al reported that surgery alone for resectable gastroesophageal cancer had the best prognosis of MSI-H or Mismatch repair (MMR) deficiency (including *MLH1* methylation) groups compared to MSI-L or MSS groups [39]. On the other hand, the prognosis of MSI-H or MMR deficiency group with chemotherapy plus surgery was poor. Therefore, MSI-H or MMR deficiency groups may be resistance to chemotherapy. An et al also found that methylated *MLH1* tended to have a better prognosis than unmethylated (MST: 68.4 vs 28.4 months, *P* = 0.49), and that CIMP-high due to multiple methylated genes including *MLH1* was CIMP-low. They reported that the prognosis was significantly better than that of CIMP-negative (*P* = 0.04) [40].

We also investigated EBV-associated gastric cancer in this study. EBV-associated gastric cancer was recognized in 7 of 138 (5.1%). This frequency is less than that of The Cancer Genome Atlas data (8.8%) [4], but recapitulates the recent large-scale report of EBV-associated gastric cancer (4.6%) [41], where EBV-associated gastric cancer was mutually exclusive of MSI-associated gastric cancer. Even in the current study, EBV-associated gastric cancer was exclusive of MSI-H/MSH-L. However, EBV-associated gastric cancer showed MSI-haplo in 2 cases (case 48 and case 103 of Fig 1A, right panel) among the 7 positive cases. EBV-associated gastric cancer almost inevitably harbored PI3 kinase mutation and p16 gene silencing [4, 41], which may be due to epigenetic carcinogenesis and therefore not completely exclusive of *MLH1* methylation in the previous literature [42].

## Conclusions

In this study we, for the first time, present critical data of clinicopathological relevance of *MLH1* hypermethylation assessed in qMSP in primary gastric cancer. We found that minute hypermethylation of *MLH1* may result in haploinsufficiency of *MLH1* expression, which represents unique phenotypes of MSI-gastric carcinogenesis. The frequency of MSI-haplo is much higher than that reported previously by the conventional MSI. As it is clinically distinct

from otherwise gastric cancer, keeping MSI-phenotype, this discovery may be beneficial for the development of therapeutic strategy for gastric cancer according to molecular classification.

## Supporting information

**S1 Fig. Methylation levels of *MLH1* promoter sequences in SW48 cells (methylation-positive control) and SW480 cells (methylation-negative control) using DNA bisulfite treatment followed by direct sequence analysis.**
(TIF)

**S2 Fig. Kaplan-Meier survival curves for OS in early (depth of invasion of pT1) gastric cancer ($P$ = 0.7904) nor in recurrent cases undergoing chemotherapy ($P$ = 0.5712).**
(TIF)

**S3 Fig. Relation between hypermethylation of *MLH1* and hypermethylation of *CDO1* (R = 0.433, $P$ < 0.0001).**
(TIF)

**S4 Fig. Direct sequencing of *MLH1* methylation in each microsatellite instability phenotype (MSI-H, MSI-L, MSS).**
(TIF)

**S1 Table. Patient's characteristics.**
(XLSX)

**S2 Table. Sequences of primers and fluorescent probes.**
(XLSX)

**S3 Table. Univariate and multivariate prognostic analysis for overall survival.**
(XLSX)

**S4 Table. Clinicopathological factors in MSI phenotypes associated with *MLH1* methylation status.**
(XLSX)

**S1 Raw images.**
(TIF)

## Acknowledgments

We thank Miss Tomomi Miyake for her technical assistance.

## Author Contributions

**Conceptualization:** Hiroki Harada, Motohiro Chuman, Yusuke Kumamoto, Takeshi Naitoh, Naoki Hiki, Keishi Yamashita.

**Data curation:** Hiroki Harada, Yusuke Nie, Ippeita Araki, Takafumi Soeno, Motohiro Chuman, Marie Washio, Mikiko Sakuraya, Hideki Ushiku, Masahiro Niihara, Kei Hosoda, Keishi Yamashita.

**Formal analysis:** Hiroki Harada, Yusuke Nie, Ippeita Araki, Takafumi Soeno, Motohiro Chuman, Marie Washio, Kei Hosoda, Yusuke Kumamoto, Keishi Yamashita.

**Funding acquisition:** Hiroki Harada, Kei Hosoda, Keishi Yamashita.

**Investigation:** Hiroki Harada, Yusuke Nie, Marie Washio, Mikiko Sakuraya, Hideki Ushiku, Naoki Hiki, Keishi Yamashita.

**Methodology:** Hiroki Harada, Motohiro Chuman, Marie Washio, Masahiro Niihara, Keishi Yamashita.

**Project administration:** Hiroki Harada, Hideki Ushiku, Takeshi Naitoh, Takafumi Sangai, Naoki Hiki, Keishi Yamashita.

**Resources:** Hiroki Harada, Takafumi Soeno, Mikiko Sakuraya, Hideki Ushiku, Kei Hosoda, Keishi Yamashita.

**Software:** Hiroki Harada, Hideki Ushiku, Masahiro Niihara, Kei Hosoda, Keishi Yamashita.

**Supervision:** Hiroki Harada, Yusuke Kumamoto, Takeshi Naitoh, Takafumi Sangai, Naoki Hiki, Keishi Yamashita.

**Validation:** Hiroki Harada, Yusuke Nie, Ippeita Araki, Takafumi Soeno, Takeshi Naitoh, Naoki Hiki, Keishi Yamashita.

**Visualization:** Hiroki Harada, Mikiko Sakuraya, Masahiro Niihara, Keishi Yamashita.

**Writing – original draft:** Hiroki Harada, Keishi Yamashita.

**Writing – review & editing:** Hiroki Harada, Yusuke Kumamoto, Takeshi Naitoh, Takafumi Sangai, Naoki Hiki, Keishi Yamashita.

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
