## [Decision Letter · Decision Letter 0]

4 Jun 2021

PONE-D-21-14491

Haploinsufficiency by minute MutL homolog 1 promoter DNA methylation may represent unique phenotypes of microsatellite instability-gastric carcinogenesis

PLOS ONE

Dear Dr. Yamashita,

Thank you for submitting your manuscript to PLOS ONE. After careful consideration, we feel that it has merit but does not fully meet PLOS ONE’s publication criteria as it currently stands. Therefore, we invite you to submit a revised version of the manuscript that addresses the points raised during the review process.

This manuscript was carefully reviewed by 2 experts, and both of them found several concerns which need to be addressed before acceptance. For instance, both reviewers require more careful explanation for the definition of haploinsufficiency of MLH1. Please respond to each of the reviewer comments.

We look forward to receiving your revised manuscript.

Kind regards,

Hiromu Suzuki, M.D., Ph.D.

Academic Editor

PLOS ONE

Journal Requirements:

2.PLOS ONE now requires that authors provide the original uncropped and unadjusted images underlying all blot or gel results reported in a submission’s figures or Supporting Information files. This policy and the journal’s other requirements for blot/gel reporting and figure preparation are described in detail at https://journals.plos.org/plosone/s/figures#loc-blot-and-gel-reporting-requirements and https://journals.plos.org/plosone/s/figures#loc-preparing-figures-from-image-files. When you submit your revised manuscript, please ensure that your figures adhere fully to these guidelines and provide the original underlying images for all blot or gel data reported in your submission. See the following link for instructions on providing the original image data: https://journals.plos.org/plosone/s/figures#loc-original-images-for-blots-and-gels.

3.Thank you for stating the following in the Acknowledgments Section of your manuscript:

"This study was

22

also supported by the Integrative 1 Research Program grant of the Graduate School of

2 Medical Science, Kitasato University and Parents’ Association Grant of Kitasato

3 University School of Medicine."

 "NO"

Reviewers' comments:

Reviewer's Responses to Questions

**Comments to the Author**

1. Is the manuscript technically sound, and do the data support the conclusions?

Reviewer #1: Yes

Reviewer #2: No

2. Has the statistical analysis been performed appropriately and rigorously? 

Reviewer #1: Yes

Reviewer #2: I Don't Know

3. Have the authors made all data underlying the findings in their manuscript fully available?

Reviewer #1: Yes

Reviewer #2: Yes

4. Is the manuscript presented in an intelligible fashion and written in standard English?

Reviewer #1: Yes

Reviewer #2: Yes

5. Review Comments to the Author

Reviewer #1: To authors:

In this study, the authors showed that haplo-MSI is asscoated with clinicopathological features of gastric cancer. Although the result of this study is interesting, there are some minor concerns.

1. The authors demonstrated that there is a subgroup that has a very low level of MLH1 methylation (MLH TaqMeth value range 0.23 to 2.23) and haplo-MSI including this subgroup is associated with clinicopathological features of gastric cancer. Are there any characteristics in this subgroup compared with MSS or other MSI groups?

2. The authors shoud showed which threshold of TaqMeth value was used for the assessment of MLH1 methylation in normal gastric mucosa in Fig 2g.

3. The definition of haplo-MSI is slightly difficult to understand, please clarify the range of MLH Taqmeth value in Fig 5f.

4. The labels of methylation level in the lower panel of Fig 1b should be showed.

Reviewer #2: In the current study, the authors investigated promoter methylation of MutL homolog 1 (MLH1) in primary gastric cancer. They revealed MLH1 hypermethylation validated in the quantitative methylation specific PCR (qMSP) assessment, was correlated with microsatellite instability (MSI)-High/MSI-Low status in MSI-associated gastric cancer. Gastric cancer patient with MLH1 hypermethylation showed better prognosis. Based on the MLH1 methylation status, MSI could be further divided into MSI-H, MSI-L, and haploinsufficiency of MSI (MSI-haplo). This study offers a molecular classification of MSI-associated tumor may be beneficial for development of therapeutic strategy for gastric cancer. Following are some comments for this study.

1. In the introduction (page 5), the authors stated that “Most of the data on MLH1 hypermethylation have been assessed by old fashioned non-quantitated conventional methylation specific PCR (MSP), and have not been properly validated in the quantitative assessment.” However, several studies demonstrated quantitative hmlh1 methylation using qMSP in gastric cancer (Taqman or methylight, PMID: 22513802, 19957042), qMSP (sybergreen, PMID: 21599969) or bisulphite pyrosequencing (PMID:17408633). Can the authors further discuss the difference between the current studies and those studies in terms of primer location and the association to clinical parameters. And why the current is superior than the others. Otherwise, it has to be rephrased.

2. The current study found that gastric cancer patient with hMLH1 hypermethylation showed better prognosis. However, several studies showed that hMLH1 methylation associated with poor prognosis in gastric cancer, for example ( PMID:29719511)

3. It is difficult for this reviewer to understand how the author define MLH1-Taqmeth value 0.23 as haploinsufficiency of MLH1 expression.

4. Morphological types should be defined as intestinal, diffuse and mixed type. Association between hMLH1 methylation with HP status is missing.

5. Clinicopathologic characteristics of MSI-associated gastric cancer needs to be delineated more in the introduction.

6. Figure 1a, is the bold line in qMSP (original) indicates the location of Taqman probe, it has to be stated in the figure legend.

7. Figure 1b, lower panel, label of Y-axis is missing.

6. PLOS authors have the option to publish the peer review history of their article (what does this mean?). If published, this will include your full peer review and any attached files.

Reviewer #1: No

Reviewer #2: No

---

## [Author Response · Author response to Decision Letter 0]

9 Jul 2021

Reviewer #1: To authors. In this study, the authors showed that haplo-MSI is asscoated with clinicopathological features of gastric cancer. Although the result of this study is interesting, there are some minor concerns.

1) The authors demonstrated that there is a subgroup that has a very low level of MLH1 methylation (MLH TaqMeth value range 0.23 to 2.23) and haplo-MSI including this subgroup is associated with clinicopathological features of gastric cancer. Are there any characteristics in this subgroup compared with MSS or other MSI groups?

 Thank you for your pertinent comments. Following Reviewer's suggestion, we examined the clinicopathological traits of True MSS, Haplo-MSI, Low-MSI, and High-MSI and identified the factors associated with their respective characteristics.

 As the MLH1 methylation value increased in the order of True MSS, Haplo-MSI, Low-MSI, and High MSI, the gastric cancers with lower tumor site (P = 0.0313), the synchronous gastric cancers (P = 0.0122) and the histologically differentiated types (P = 0.0052) increased (Table S4).

 The above contents have been added to Table S4 and Results (P17, Line7-10).

2) The authors should show which threshold of TaqMeth value was used for the assessment of MLH1 methylation in normal gastric mucosa in Fig 2g.

 Thank you for your pertinent comments. From the viewpoint of field cancerization, we initially supposed that MLH1 methylation may also be observed in both the synchronous tumor and corresponding normal mucosa as well as in the main tumor. In fact, however, this was not true, and not all mucosa harbored hypermethylation of the MLH1 and methylation patterns were rather random. Thus, threshold was not set to discriminate the corresponding normal mucosa from tumor tissues.

 The above contents have been added to Results (P13, Line7-8).

3) The definition of haplo-MSI is slightly difficult to understand, please clarify the range of MLH Taqmeth value in Fig 5f.

 Thank you for your appropriate suggestions. We are considering that haplo-MSI phenotype could be microsatellite-instable despite MSS defined by the current definition. The definition of MSI-H/L is based on the specific relevant markers (Bethesda panel), and they could not be covered with minor abnormalities of microsatellites caused by haplo-MSI that is corresponding to the lowest hypermethylation of MLH1 (> 0.23) (Fig. 5f). Prognostic stratification defined by the most optimal cut-off value (0.23, but not 2.3) might propose bona-fide entity of MSI phenotypes including haplo-MSI in gastric cancer. 

 The position of the MLH1-Taqmeth value in Figure 5f has been changed to clarify the range of each MSI phenotype. The above contents have been added to Discussion (P19, Line15-21).

4) The labels of methylation level in the lower panel of Fig 1b should be showed.

 Thank you for your appropriate suggestions. A label for the methylation level is displayed on the Y-axis in the lower panel of Figure 1b.

Reviewer #2: In the current study, the authors investigated promoter methylation of MutL homolog 1 (MLH1) in primary gastric cancer. They revealed MLH1 hypermethylation validated in the quantitative methylation specific PCR (qMSP) assessment, was correlated with microsatellite instability (MSI)-High/MSI-Low status in MSI-associated gastric cancer. Gastric cancer patient with MLH1 hypermethylation showed better prognosis. Based on the MLH1 methylation status, MSI could be further divided into MSI-H, MSI-L, and haploinsufficiency of MSI (MSI-haplo). This study offers a molecular classification of MSI-associated tumor may be beneficial for development of therapeutic strategy for gastric cancer. Following are some comments for this study.

1) In the introduction (page 5), the authors stated that “Most of the data on MLH1 hypermethylation have been assessed by old fashioned non-quantitated conventional methylation specific PCR (MSP), and have not been properly validated in the quantitative assessment.” However, several studies demonstrated quantitative hmlh1 methylation using qMSP in gastric cancer (Taqman or methylight, PMID: 22513802, 19957042), qMSP (sybergreen, PMID: 21599969) or bisulphite pyrosequencing (PMID:17408633). Can the authors further discuss the difference between the current studies and those studies in terms of primer location and the association to clinical parameters. And why the current is superior than the others. Otherwise, it has to be rephrased.

 Thank you for providing these insights. Both reports using Taqman or MethyLight (PMID 22513802, 19957042) examined quantitative hMLH1 methylation by qMSP in gastric cancer, but did not clarify its clinicopathological relevance including prognostic factors such as TNM. On the other hand, our current study quantified MLH1 methylation of primary gastric tumors by qMSP using Taqman probe and evaluated its clinical pathological relations. 

 Moreover, we for the first time proposed that quantitative values in MLH1 methylation may have a gradual effect on the MSI phenotype and prognosis in Q-MSP based on studies which originally defined positive control and negative control by cloned sequence. Therefore, our current study is considered to be unique for investigating clinicopathological factors using quantitative values of MLH1 methylation in gastric cancer. 

 However, as the reviewers pointed out, "Most of the data on MLH1 hypermethylation have been assessed by old-fashioned non-quantitated conventional methylation specific PCR (MSP), and have not been properly validated in the quantitative assessment." The phrase was not appropriate. "Most of the data on MLH1 hypermethylation were evaluated by non-quantitative methylation-specific PCR (MSP). In previous similar reports evaluated by quantitative MSP, the quantified methylation values were not applied to investigate clinicopathological factors." We changed it to the phrase.

 The above contents have been added to Introduction (P5, Line3-6).

2) The current study found that gastric cancer patient with hMLH1 hypermethylation showed better prognosis. However, several studies showed that hMLH1 methylation associated with poor prognosis in gastric cancer, for example (PMID:29719511)

 Thank you for your pertinent comments. We have confirmed the report of the meta-analysis pointed out by the reviewers (Peng Y. Front Physiol 2018). It was reported that hMLH1 methylation in gastric cancer has a considerable association in lymph node metastasis, microsatellite status, Lauren classification, Helicobacter pylori infection, and hMLH1 protein expression. On the other hand, we could not find the results regarding the prognosis in the manuscript. 

 We also reviewed all reports used to analyze lymph node metastases that may the most strongly affect prognosis, but none reported prognostic analysis (Ferrasi AC. World J Gastroenterol 2010, Hiraki). M. An Surg Oncol 2010, Jin J. Genet Mol Res 2014, Kolesnikova EV. Ann NY Acad Sci 2008, Moghbeli M. J Gastrointest Cancer 2014, Nakajima T. Int J Cancer 2001, Pinto M. Lab Invest 2000, Song B. Pak J Med Sci 2013, Xiong HL. Asian Pac J Cancer 2013).

 Smyth et al reported that surgery alone for resectable gastroesophageal cancer had the best prognosis of MSI-H or Mismatch repair (MMR) deficiency (including MLH1 methylation) groups compared to MSI-L or MSS groups (Smyth EC. JAMA Oncol 2017). On the other hand, the prognosis of MSI-H or MMR deficiency group with chemotherapy plus surgery was poor. Therefore, MSI-H or MMR deficiency groups may be resistance to chemotherapy. An et al also found that methylated MLH1 tended to have a better prognosis than unmethylated (MST: 68.4 vs 28.4 months, P = 0.49), and that CIMP-high due to multiple methylated genes including MLH1 was CIMP-low. They reported that the prognosis was significantly better than that of CIMP-negative (P = 0.04) (An C. Clin Cancer Res 2005).

 Based on the above reports, hypermethylation of MLH1 may be involved in chemotherapy resistance, but there are no reports that it affects the poor prognosis. Therefore, the prognostic analysis results of this study were considered to be more acceptable allowing for the previous reports.

 The above contents have been added to Discussion (P21, Line13-21).

3) It is difficult for this reviewer to understand how the author define MLH1-Taqmeth value 0.23 as haploinsufficiency of MLH1 expression.

 Thank you for your very important point. We are considering that haplo-MSI phenotype could be microsatellite-instable despite MSS defined by the current definition. The definition of MSI-H/L is based on the specific relevant markers (Bethesda panel), and they could not be covered with minor abnormalities of microsatellites caused by haplo-MSI that is corresponding to the lowest hypermethylation of MLH1 (> 0.23) (Fig. 5f). Prognostic stratification defined by the most optimal cut-off value (0.23, but not 2.3) might propose bona-fide entity of MSI phenotypes including haplo-MSI in gastric cancer. 

 The position of the MLH1-Taqmeth value in Figure 5f has been changed to clarify the range of each MSI phenotype. The above contents have been added to Discussion (P19, Line15-21).

4) Morphological types should be defined as intestinal, diffuse and mixed type. Association between hMLH1 methylation with HP status is missing.

 Thank you for your suggestions. The Lauren classification is considered to be a classification method that divides the histological, but not morphological, classification in gastric cancer into intestinal, diffuse, and mixed types (Lauren P. Acta Pathol Microbiol Scand. 1965; 64: 31-49.).

5) Clinicopathologic characteristics of MSI-associated gastric cancer needs to be delineated more in the introduction.

 Thank you for your pertinent comments. It has been reported that the clinicopathological factors in the MSI-associated gastric cancer were elderly, female, intestinal type in Lauren classification, and tumors located in the middle to lower site of the stomach (TCGA. Nature 2014).

 The above contents have been added to Introduction (P4, Line13-15).

6) Figure 1a, is the bold line in qMSP (original) indicates the location of Taqman probe, it has to be stated in the figure legend.

 Thank you for your appropriate suggestions. As you pointed out, the bold line in qMSP (original) indicates the location of the Taqman probe.

 The above contents have been added to Figure Legends (P29, Line6-7).

7) Figure 1b, lower panel, label of Y-axis is missing.

 Thank you for your pertinent comments. A label for the methylation level is displayed on the Y-axis in the lower panel of Figure 1b.

---

## [Decision Letter · Decision Letter 1]

4 Aug 2021

PONE-D-21-14491R1

Haploinsufficiency by minute MutL homolog 1 promoter DNA methylation may represent unique phenotypes of microsatellite instability-gastric carcinogenesis

PLOS ONE

Dear Dr. Yamashita,

Thank you for submitting your manuscript to PLOS ONE. After careful consideration, we feel that it has merit but does not fully meet PLOS ONE’s publication criteria as it currently stands. Therefore, we invite you to submit a revised version of the manuscript that addresses the points raised during the review process.

The authors addressed most of the issues raised by the editors. However, reviewer 2 suggests an additional experiment to ensure the reliability of qMSP. Please respond to the reviewer comment.

We look forward to receiving your revised manuscript.

Kind regards,

Hiromu Suzuki, M.D., Ph.D.

Academic Editor

PLOS ONE

Journal Requirements:

Reviewers' comments:

Reviewer's Responses to Questions

**Comments to the Author**

1. If the authors have adequately addressed your comments raised in a previous round of review and you feel that this manuscript is now acceptable for publication, you may indicate that here to bypass the “Comments to the Author” section, enter your conflict of interest statement in the “Confidential to Editor” section, and submit your "Accept" recommendation.

Reviewer #1: All comments have been addressed

Reviewer #2: (No Response)

2. Is the manuscript technically sound, and do the data support the conclusions?

Reviewer #1: Yes

Reviewer #2: Yes

3. Has the statistical analysis been performed appropriately and rigorously? 

Reviewer #1: Yes

Reviewer #2: Yes

4. Have the authors made all data underlying the findings in their manuscript fully available?

Reviewer #1: Yes

Reviewer #2: Yes

5. Is the manuscript presented in an intelligible fashion and written in standard English?

Reviewer #1: Yes

Reviewer #2: Yes

6. Review Comments to the Author

Reviewer #1: (No Response)

Reviewer #2: The authors have addressed most of the comments. However, to avoid false positivity of qMSP, bisulphite sequencing of hMHL1 should be performed in selective samples in the haplo-MSI and low/high-MSI group

7. PLOS authors have the option to publish the peer review history of their article (what does this mean?). If published, this will include your full peer review and any attached files.

Reviewer #1: No

Reviewer #2: No

---

## [Author Response · Author response to Decision Letter 1]

24 Oct 2021

PLoS One

October 14, 2021

Dear Prof. Hiromu Suzuki, Editor

Thank you for inviting us to submit a revised draft of our manuscript entitled, “Haploinsufficiency by minute MutL homolog 1 promoter DNA methylation may represent unique phenotypes of microsatellite instability-gastric carcinogenesis” to PLoS One. We also appreciate the time and effort you and each of the reviewers have dedicated to providing insightful feedback on ways to strengthen our paper. Thus, it is with great pleasure that we resubmit our article for further consideration. We have incorporated changes that reflect the detailed suggestions you have graciously provided. We also hope that our edits and the responses we provide below satisfactorily address all the issues and concerns you and the reviewers have noted.

To facilitate your review of our revisions, the following is a point-by-point response to the questions and comments delivered in your letter.

Reviewer's Responses to Questions

Comments to the Author

1. If the authors have adequately addressed your comments raised in a previous round of review and you feel that this manuscript is now acceptable for publication, you may indicate that here to bypass the “Comments to the Author” section, enter your conflict-of-interest statement in the “Confidential to Editor” section, and submit your "Accept" recommendation.

Reviewer #1: All comments have been addressed

Reviewer #2: (No Response)

2. Is the manuscript technically sound, and do the data support the conclusions?

Reviewer #1: Yes

Reviewer #2: Yes

3. Has the statistical analysis been performed appropriately and rigorously?

Reviewer #1: Yes

Reviewer #2: Yes

4. Have the authors made all data underlying the findings in their manuscript fully available?

Reviewer #1: Yes

Reviewer #2: Yes

5. Is the manuscript presented in an intelligible fashion and written in standard English?

Reviewer #1: Yes

Reviewer #2: Yes

6. Review Comments to the Author

Reviewer #1: (No Response)

Reviewer #2: The authors have addressed most of the comments. However, to avoid false positivity of qMSP, bisulphite sequencing of hMHL1 should be performed in selective samples in the haplo-MSI and low/high-MSI group

Thank you for your pertinent comments. Following reviewer's suggestion, we examined eight cases using direct sequencing. As a result, methylation could be detected in only 3 cases. The remaining 5 cases were MSI-L or MSS cases. Therefore, it was considered that the sensitivity to detect methylation of direct sequencing is limited to MLH1 hypermethylation cases, and MSI-haplo can only be detected by real-time PCR. The above contents have been added to Table and Discussion (P19, Line23-P20, Line 3).

7. PLOS authors have the option to publish the peer review history of their article (what does this mean?). If published, this will include your full peer review and any attached files.

Do you want your identity to be public for this peer review? For information about this choice, including consent withdrawal, please see our Privacy Policy.

Reviewer #1: No

Reviewer #2: No

In response to this revision, we asked the following doctor to give advice on the experiment, analysis and correct the rebuttal letter. Therefore, we would like to add the following one as co-author.

Yusuke Nie 2, Department of General-Pediatric-Hepatobiliary Pancreatic Surgery, Kitasato University School of Medicine, 1-15-1 Kitasato, Minami-ku, Sagamihara, Kanagawa 252-0375, Japan

Ippeita Araki 1, Department of Upper-gastrointestinal Surgery, Kitasato University School of Medicine, 1-15-1 Kitasato, Minami-ku, Sagamihara, Kanagawa 252-0375, Japan

We have found some typographical errors in the base sequence of Table S2. Therefore, we corrected it.

Again, thank you for giving us the opportunity to strengthen our manuscript with your valuable comments and queries. We have worked hard to incorporate your feedback and hope that these revisions persuade you to accept our submission.

Sincerely,

Hiroki Harada, MD, PhD

Keishi Yamashita, MD, PhD, FACS

Corresponding Author: Keishi Yamashita

Division of Advanced Surgical Oncology, Department of Research and Development Center for New Medical Frontiers, Kitasato University School of Medicine

1-15-1, Kitasato, Minami-ku, Sagamihara, KANAGAWA, 252-0374, JAPAN

E-mail: keishi23@med.kitasato-u.ac.jp

Tel: +81-42-778-8111 / Fax: +81-42-778-9556

---

## [Decision Letter · Decision Letter 2]

8 Nov 2021

Haploinsufficiency by minute MutL homolog 1 promoter DNA methylation may represent unique phenotypes of microsatellite instability-gastric carcinogenesis

PONE-D-21-14491R2

Dear Dr. Yamashita,

We’re pleased to inform you that your manuscript has been judged scientifically suitable for publication and will be formally accepted for publication once it meets all outstanding technical requirements.

Kind regards,

Hiromu Suzuki, M.D., Ph.D.

Academic Editor

PLOS ONE

Additional Editor Comments (optional):

Reviewers' comments:

Reviewer's Responses to Questions

**Comments to the Author**

1. If the authors have adequately addressed your comments raised in a previous round of review and you feel that this manuscript is now acceptable for publication, you may indicate that here to bypass the “Comments to the Author” section, enter your conflict of interest statement in the “Confidential to Editor” section, and submit your "Accept" recommendation.

Reviewer #1: All comments have been addressed

Reviewer #2: All comments have been addressed

2. Is the manuscript technically sound, and do the data support the conclusions?

Reviewer #1: Yes

Reviewer #2: Yes

3. Has the statistical analysis been performed appropriately and rigorously? 

Reviewer #1: Yes

Reviewer #2: I Don't Know

4. Have the authors made all data underlying the findings in their manuscript fully available?

Reviewer #1: Yes

Reviewer #2: Yes

5. Is the manuscript presented in an intelligible fashion and written in standard English?

Reviewer #1: Yes

Reviewer #2: Yes

6. Review Comments to the Author

Reviewer #1: The authors addressed all data adequately, and the results were well discussed.

So I have no concern about publishing this paper.

Reviewer #2: (No Response)

7. PLOS authors have the option to publish the peer review history of their article (what does this mean?). If published, this will include your full peer review and any attached files.

Reviewer #1: No

Reviewer #2: No

---

## [Editor Report · Acceptance letter]

13 Dec 2021

PONE-D-21-14491R2 

Haploinsufficiency by minute *MutL homolog 1* promoter DNA methylation may represent unique phenotypes of microsatellite instability-gastric carcinogenesis 

Dear Dr. Yamashita:

I'm pleased to inform you that your manuscript has been deemed suitable for publication in PLOS ONE. Congratulations! Your manuscript is now with our production department. 

Kind regards, 

on behalf of

Dr. Hiromu Suzuki 

Academic Editor

PLOS ONE